# Development of an Interference Filter-Stabilized External-Cavity Diode Laser for Space Applications

**Linbo Zhang** [1,2,3] 📷, **Tao Liu** [1,*], **Long Chen** [1,3], **Guanjun Xu** [1], **Chenhui Jiang** [1,3] **and Jun Liu** [1] **and Shougang Zhang** [1]

1   Key Laboratory of Time and Frequency Primary Standards, National Time Service Center, Chinese Academy of Sciences, Xi'An 710600, China; linbo@ntsc.ac.cn (L.Z.); chenlong@ntsc.ac.cn (L.C.); xuguanjun@ntsc.ac.cn (G.X.); jiangchenhui15@mails.ucas.edu.cn (C.J.); liujun@ntsc.ac.cn (J.L.); szhang@ntsc.ac.cn (S.Z.)
2   State Key Laboratory of Transient Optics and Photonics, Chinese Academy of Sciences, Xi'An 710600, China
3   University of Chinese Academy of Sciences, Beijing 100049, China
*   Correspondence: taoliu@ntsc.ac.cn; Tel.: +86-29-8389-1031

**Abstract:** The National Time Service Center of China is developing a compact, highly stable, 698 nm external-cavity diode laser (ECDL) for dedicated use in a space strontium optical clock. This article presents the optical design, structural design, and preliminary performance of this ECDL. The ECDL uses a narrow-bandwidth interference filter for spectral selection and a cat's-eye reflector for light feedback. To ensure long-term stable laser operation suitable for space applications, the connections among all the components are rigid and the design avoids any spring-loaded adjustment. The frequency of the first lateral rocking eigenmode is 2316 Hz. The ECDL operates near 698.45 nm, and it has a current-controlled tuning range over 40 GHz and a PZT-controlled tuning range of 3 GHz. The linewidth measured by the heterodyne beating between the ECDL and an ultra-stable laser with 1 Hz linewidth is about 180 kHz. At present, the ECDL has been applied to the principle prototype of the space ultra-stable laser system.

**Keywords:** external-cavity diode laser; interference filter; laser diode; laser stabilization; space optical clock

## 1. Introduction

Compared to most other types of laser, diode lasers are cheap and simple to use; they also have a high power and cover a large wavelength range. They have therefore become attractive light sources with versatile applications in many fields of optical technology and experimental physics, such as optical atomic clocks, precision measurement, astrophysics, and quantum communication [1–5]. With this wide spectrum of applications, it is not surprising that lasers are used in very different environments, with one of the most demanding being space [6].

In 1980, Lang and Kobayashi [7] applied external-cavity feedback technology to diode lasers. The increased external cavity can narrow the laser line width, and The wavelength can be tuned by changing the external cavity length. In The following years, Soviet scientists, for The first time, used a diffraction grating to feed back part of the output light of the diode laser to the active region, narrowing the linewidth of the laser to 1.5 MHz [8]. Common external-cavity diode laser (ECDL) designs such as the Littrow [9,10] and Littman–Metcalf configurations [11,12] use diffraction gratings for wavelength selection. Those lasers require precise alignment and are therefore sensitive to acoustic and mechanical disturbances, particularly when a spring-loaded kinematic mount is used to align the grating or feedback optics [13]. Another design uses a narrowband interference filter (IF) placed in a

linear cavity as a frequency-selective component [14–16]. Because The wavelength of the transmission maximum depends on the angle of incidence, where the angle is the angle of incidence and the maximum wavelength is the transmitted wavelength at normal incidence, the wavelength can be tuned by turning the filter. This leads to a sensitivity of $d\lambda/d\theta = 0.017$ nm/mrad, which is 60 times better than that of the Littrow configuration. Thus, the laser design is in principle less sensitive to mechanical vibrations and disturbances. With these advantages, the interference-filter configuration was chosen for the PHARAO [17,18] and SOC2 [19] projects for space laser systems.

As the atomic clock with the highest performance index in the world, the measurement accuracy of a strontium atomic optical clock has entered the order of $10^{-19}$ [20]. In the microgravity environment of space, the performance of optical clocks is expected to be further improved [21]. The National Time Service Center (NTSC) of China is conducting research on the space Sr atomic optical clock. In the strontium atomic optical clock system, the wavelength of the clock transition $^1S_0 \rightarrow^3 P_0$ is 698 nm, and The natural linewidth is only 1 mHz. The linewidth of the detection light must reach the order of Hz or even sub-Hz. We use an ultra-narrow linewidth laser to detect the clock transition line. The ultra-narrow linewidth laser (also called clock laser) was obtained by locking the laser frequency to a high-finesse optical reference cavity by means of the Pound–Drever–Hall (PDH) technique [22]. The ECDL developed in this paper is used as the light source of an ultra-narrow linewidth laser system which is aimed to has a free-run linewidth at the level of 500 kHz or less. At present, commercial semiconductor lasers at 698 nm have wavelength tuning capabilities up to 10 nm. However, due to the use of an elastically loaded adjustment device, the structure is not very stable and usually needs to be readjusted every month to ensure good optical feedback and correct wavelength. Although The laser developed in this paper does not have a wide range of tuning capabilities, it has a stable structure and is one of the best choices to meet our special applications.

The objective of the present study is to develop and characterize a prototype of a 698 nm interference-filter external-cavity diode laser (IF-ECDL). The developed ECDL is compact and robust, and it will be planned for use in China's space Sr atomic light clock system in the future.

## 2. Working Principle of the IF-ECDL

In the IF-ECDL, the interference filter provides wavelength selection, and a partially reflective mirror provides optical feedback, as shown schematically in Figure 1. The interference filter is composed of alternating layers of dielectric material that can transmit a narrow frequency band while reflecting the light of other wavelengths. The narrowband interference filter is actually a thin Fabry–Perot etalon with only one transmission peak in the visible range, also known as a line filter. The bandpass section of an interference filter is made by the repetitive vacuum deposition of thin layers of partially reflecting dielectric compounds on a glass substrate. Dielectric layers are arranged to form reflective cavities. The spacer region is designed to be $\lambda_0/2$ thick, where $\lambda_0$ is the central wavelength of the filter. This allows light that meets the reflection boundary conditions to be reinforced and transmitted by the cavity. The rejected light is reflected by the layers of dielectric material. The laser light generated by the semiconductor laser is collimated into parallel light by a lens, and then incident on the interference filter at a certain angle $\theta$. Using the multi-beam isotropic interference theory, the wavelength $\lambda$ at the peak of the transmittance is

$$\lambda = \lambda_{max}\sqrt{1 - \frac{sin^2\theta}{n_{IF}^2}} \tag{1}$$

where $n_{IF}$ is the interference filter's effective index of refraction, and $\lambda_{max}$ is the transmission wavelength value of the narrow-band filter when the beam is normally incident, and is also the maximum limit wavelength value in the tuning range of the narrow-band filter.

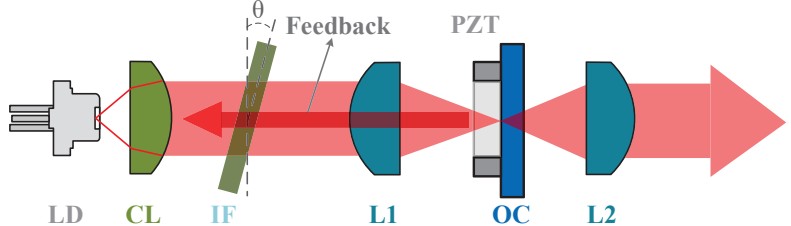

**Figure 1.** Schematic of interference-filter external-cavity diode laser (IF-ECDL). LD: laser diode; CL: collimating lens; IF: interference filter; L1: cat's-eye lens; PZT: piezotube; OC: partially reflective out-couple mirror; L2: re-collimating lens.

Assuming that the outgoing light intensity of the bare laser diode is *I*, then the total light intensity transmitted through the interference filter is given by the Airy formula

$$I_T = I_0 \frac{1}{1 + F_{IF} sin^2(\theta/2)} \tag{2}$$

where $F_{IF}$ is the fineness of the IF. By rotating the angle of the interference filter placed in the ECDL resonator cavity with respect to the laser, the wavelength of the laser exiting the IF-ECDL is tuned. Using the IF for frequency selection, a single longitudinal mode laser can be obtained.

## 3. Design of the IF-ECDL

### 3.1. Optical Design

A semiconductor laser diode (LD) with a central wavelength of 698 nm is anti-reflection coated on its output facet and is combined with an external cavity, leading to a large tuning range for the wavelength. The cavity length of the LD is 750 μm, and The reflectance of the two surfaces are 1 for the back face and $3 \times 10^{-4}$ for the AR coated face. The light coming from the LD is collimated by a collimating lens (CL) with a focal length of 4.02 mm and a numerical aperture of 0.6. To collect as much light from the diode as possible, the CL must have a large numerical aperture.

The optical feedback is provided by a combination of a cat's-eye lens (L1) with a focal length of 15.29 mm and a partially reflective out-couple mirror (OC) with 30% reflectivity in focal distance. This cat's-eye configuration is less sensitive to misalignments of the OC compared to the case of feedback with no such lens. A cat's-eye reflector decreases the sensitivity to optical misalignment and maximizes the feedback efficiency. The overall external-cavity length from the LD output facet to the OC front facet is 50 mm and corresponds to an axial mode spacing of

$$\Delta_{FSR} = \frac{c}{2L} = 3 \, GHz. \tag{3}$$

The optical length of the external cavity is tuned with a piezotube (PZT) of 9 mm in length and with internal and external diameters of 5 and 10 mm, respectively. Applying a voltage of 100 V to the PZT changes the length of the external cavity by 1.4 μm. For a given optical mode, a variation Δ*l* in the optical path *l* of the external cavity yields a relative frequency detuning of

$$\frac{\Delta \nu}{\nu} = -\frac{\Delta l}{l} \tag{4}$$

allowing the PZT to tune the laser frequency with a response of −120 MHz/V.

The final optical component in the optical path is a re-collimating lens (L2) with a focal length of 11 mm, which is used to re-collimate the out-coupled laser. To narrow the output beam, the L2 focal length is chosen to be smaller than the L1 focal length.

An IF is placed inside the external cavity between the CL and L1 and is used for coarse wavelength tuning. It is part of a resonator that forces the laser to maintain a stable single mode and reduces the linewidth. The IF is made of a substrate that is coated with many dielectric layers on one side and anti-refection coated on the other side [23]. It has a 0.48 nm super-narrow passband and a peak transmission of 96%, and its measured spectrum is shown in Figure 2. The wavelength of the transmitted light is changed by adjusting the angle of the IF. Compared with the Littrow and Littman–Metcalf configurations, the IF and cat's-eye reflector replace the grating used to select the laser wavelength and form an external cavity, making it relatively easily to adjust the laser frequency and optimize the optical feedback. Furthermore, because The IF and cat's-eye reflector are insensitive to the incident angle [15], the present design has a higher mechanical stability than those of the Littrow and Littman–Metcalf configurations.

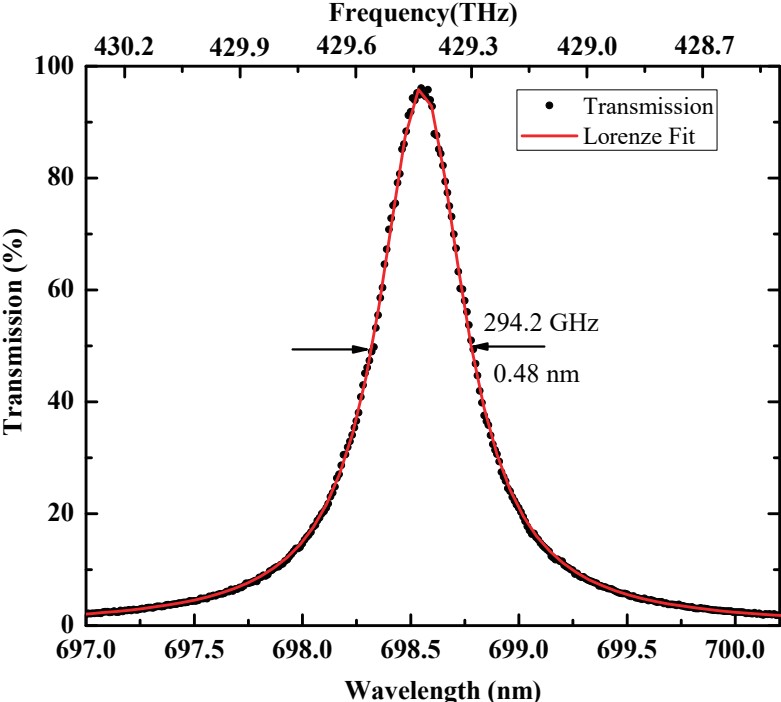

**Figure 2.** Measured transmission spectrum of interference filter (IF) with 6° angle of incidence. Data provided by manufacturer (Alluxa, Santa Rosa, CA, USA). The IF has a peak transmittance of up to 96% and a bandwidth of ∼0.48 nm(294.2 GHz).

*3.2. Structural Design*

The aim of the structural design of the ECDL is to provide a good mechanical environment for the optical components installed inside. The design should be able to resist external mechanical inputs, thereby ensuring the reliability and stability of the laser. Structural factors such as structural stability, machining and assembly accuracy, mechanical robustness, weight, and size should be considered in the design process.

The mechanical structure of the ECDL is shown in Figure 3. The mechanical parts comprise a laser base, a mount for the LD and CL, a mount for L1 and the PZT, a mount for L2, and a mount for the IF. In The present design, most of the parts are made of aluminum alloy, which is relatively light and has a consistent rate of thermal expansion, thereby reducing the effects of thermal stress on the optical components. The choice of material in this design was made primarily for principle verification; other material options that satisfy environmental requirements include AlSiC and Ti.

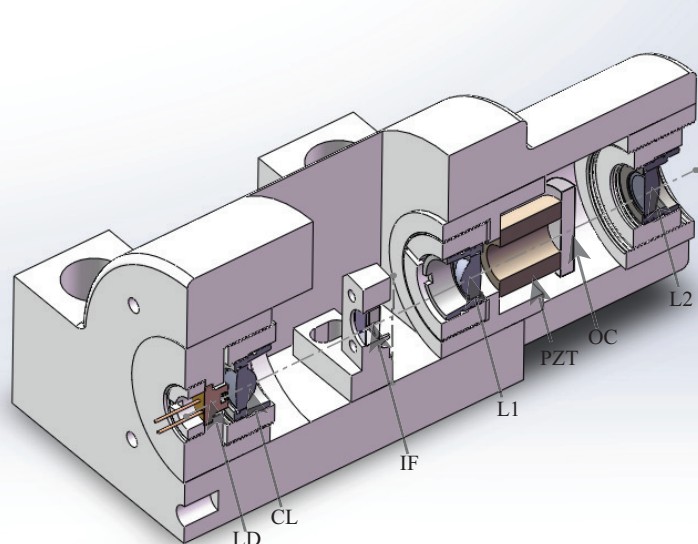

**Figure 3.** Computer-aided design view of the ECDL shown schematically in Figure 1. A sectional view is presented here to reveal the internal structure.

To ensure long-term stable laser operation suitable for space applications, the connections among all the components are rigid, and The design avoids any spring-loaded adjustment. The LD is fixed rigidly to its bracket by a retaining ring. Except for the IF, the mounts of the other optical components are cylindrical structures with the same outer diameter. All the mounts are inserted into the laser base after the components are mounted and locked by M2.5 screws. After setting the required angle of the IF holder, it is fixed to the platform of the laser base by two M3 screws. All the lenses, including CL, L1, and L2, are adjustable along the optical axis only; they are adjusted to their required positions and then fixed in place using slow-setting glue.

The laser base is machined from a solid aluminum block to ensure that the laser is stable, robust, and insensitive to outside interference. Figure 4 shows a photograph of the laser, the outer envelope of which is 75 mm × 65 mm × 39 mm. Because The laser frequency depends on the length of the cavity, precise temperature control of the laser is necessary [9,24]. A small hole with a diameter of 3 mm and a depth of 5 mm is found at the end of the laser base close to the LD; in this hole is placed a negative-temperature-coefficient thermistor for detecting variations in temperature of the LD. A Peltier thermoelectric cooler with dimensions of 40 mm × 40 mm × 4 mm is attached to the surface of the laser base to stabilize the temperature of the cavity and LD. To ensure a laser output height of 20 mm from the optical table, this ensemble is fixed on the optical platform with four M4 Teflon screws. The optical platform acts as a heat sink, as illustrated in Figure 4.

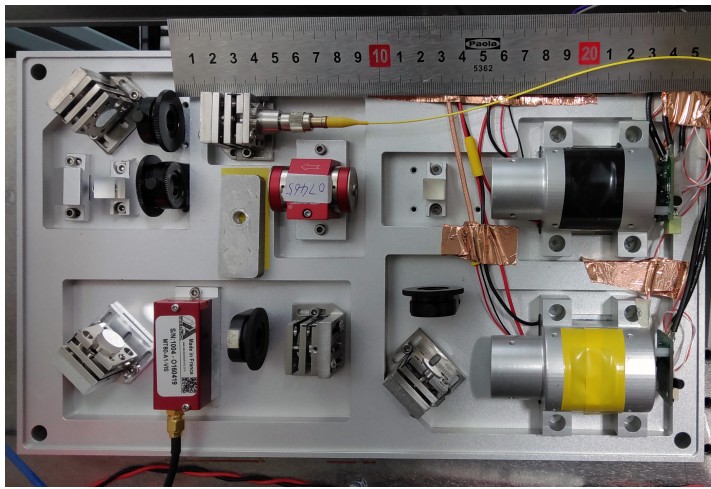

**Figure 4.** Photograph of the ECDL, which has been applied to space narrow-linewidth-laser demonstration systems.

### 3.3. Simulation of Eigenmodes and Stress Distribution

The structural design of the laser should have a high fundamental frequency and good dynamic characteristics to prevent structural damage caused by resonance of the low-frequency coupling during launch. We used finite element modeling to verify that the ECDL design is satisfactory. To reduce the amount of data needed for analog operation, we simplified the laser model appropriately. The model retains the physical structure of the laser, the optics, and The mounting brackets, but omits details such as chamfers and fillets. The overall structural material of the laser is aluminum alloy, the optical lens material is fused silica, and The PZT material is PZT-5A; the properties of these materials are given in Table 1. The simulation results are shown in Figure 5, where the lateral rocking frequency of the first eigenmode is 2316 Hz. Because The first-order natural frequency of the module at the aerospace standard component level exceeds 70 Hz, the modal analysis shows that the design meets the requirements. However, the eigenfrequency is too high, indicating that the structural design of the laser requires further optimization.

**Table 1.** Material properties.

| Material | $\rho$ (kg/m$^3$) | $\sigma_p$ | $E_{YM}(10^9$ Pa) |
|---|---|---|---|
| Aluminum 6063 | 2700 | 0.33 | 69 |
| Fused silica | 2203 | 0.17 | 73.1 |
| PZT-5A | 7750 | 0.31 | 53 |

Another issue is that the gravitational environment differs between space and the laboratory. The lab-mounted laser undergoes a tiny deformation once in microgravity. To reduce this effect, we adopted an integrated external cavity structure to reduce the relative mechanical deformation as much as possible. In addition, no adjustable elastic mechanical structure was used. The cantilever length and mass are reduced as much as possible while maintaining the mechanical strength. Figure 6 shows the displacement of the ECDL under a vertical acceleration of 1 g. The maximum deformation clearly occurs at L2 but is only 0.01 µm. When the optical board with the laser mounted is placed face up and back up, the performance of the optical system remains the same, indicating that this small deformation has no effect on the performance of the ECDL. We also analyze the deformation of two major components, IF and OC, that affect ECDL performance under the action of gravity along the optical axis (z axis). Among them, the deformation of IF is 0.003 µm. This deformation is mainly a translation in the direction of light transmission and has no effect on the angle of the IF. The deformation of OC is 0.005 µm. This deformation will increase the length of the external cavity

and affect the frequency of the output laser. This slight shift results in a frequency change of only 0.4 GHz. This deviation can be corrected by adjusting the voltage of the PZT.

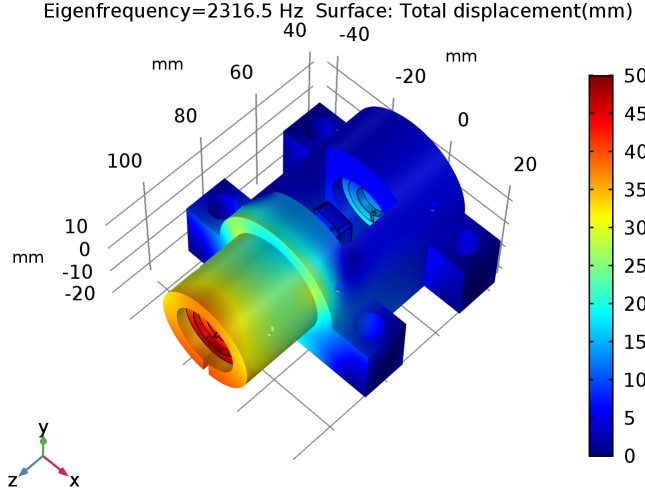

**Figure 5.** The frequency of the first lateral rocking eigenmode is 2316 Hz.

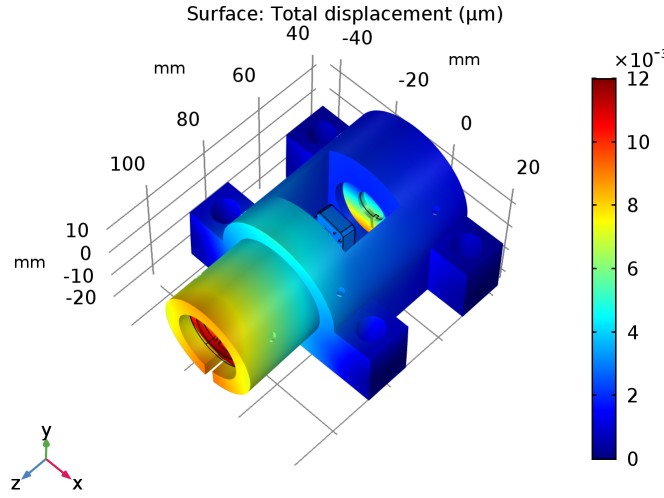

**Figure 6.** Displacement of the ECDL under action of gravitational field.

## 4. Test Results

After adding the external cavity to the diode laser, iterative focal adjustment and external-cavity alignment were implemented to optimize the optical feedback. Optimum alignment was accomplished when the threshold was reduced to a minimum [25]. The variation of output power with laser diode current as measured using an optical powermeter is shown in Figure 7 both with and without an external cavity. The threshold current of the ECDL is 30 mA, and The diode current shifts the laser output power by 0.91 mW/mA. As can be seen in Figure 7, the threshold current was reduced by approximately 10 mA compared to the bare tube, and The output optical power was increased by 15 mW at a laser current of 65 mA. The output surface of the laser diode we used was coated with an anti-reflection coating, and The reflectivity was only $10^{-4}$ orders of magnitude. In principle, the main reason for increasing the output laser power after increasing the external cavity was to reduce the threshold current: (1) adding an external cavity is equivalent to an increase in cavity length; (2) introducing optical feedback to help increase the stimulated emission suppresses the spontaneous

radiation. These results indicate that the external-cavity semiconductor laser of the present design achieves strong optical feedback and completes good alignment [25].

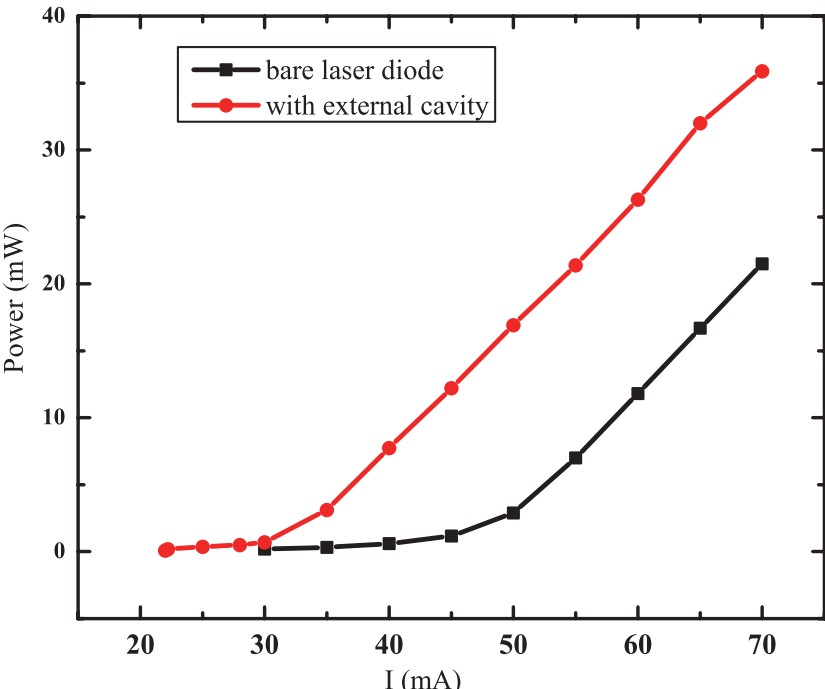

**Figure 7.** Output power versus laser diode current as measured using an optical powermeter at a controlled laser temperature of 22.3 °C.

The effect of injection current and temperature on the output wavelength of the ECDL was investigated. It can be seen in Figure 8 that, like the laser diode, the ECDL also has a mode hopping interval. It is necessary to avoid this mode hopping interval when the laser is working. The wavelength dependence on the injection current at a fixed temperature of 22.3 °C is shown in Figure 8a. The current adjustment range corresponding to the ECDL's no mode hopping interval is about 9 mA, and the corresponding frequency varies by approximately 43 GHz. From this, the frequency tuning rate of the injection current can be calculated to be approximately 4.8 GHz/mA. Figure 8b shows the wavelength dependence on temperature at a fixed current of 64 mA. The coefficient of the frequency with temperature is 25 GHz/°C, and The non-mode hopping interval is 1.7 °C. The laser frequency can be tuned over a wide range by changing the temperature, but this adjustment is very rough, and it takes a long time for the laser temperature to become completely stable. With increasing injection current or temperature, the wavelength increased (frequency decreased). This is because the temperature of the laser diode increases as the injection current increases. The effective refractive index increased, leading to an increase in the optical length of the internal cavity. When the wavelength increased and entered the edge of the range selected by the interference filter, the operating mode competed with neighboring modes and mode hopping occurred. The range of continuous non-hopping mode is mainly determined by the FSR of the internal cavity and the full width at half maximum (FWHM) of the interference filter.

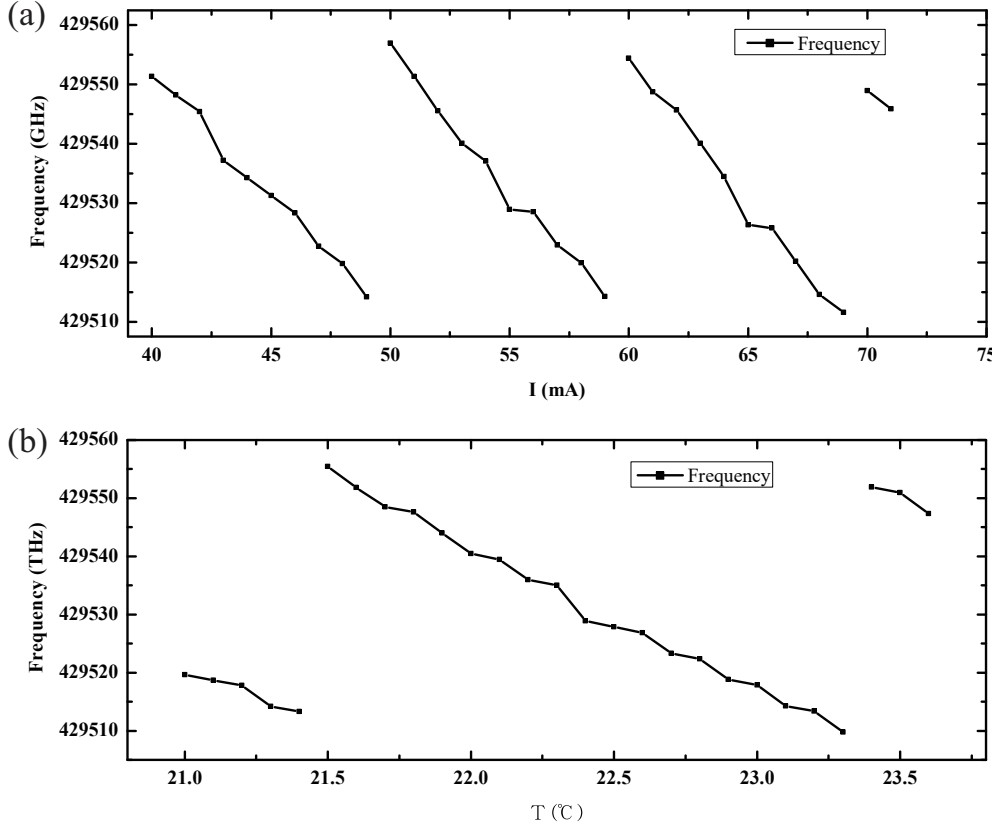

**Figure 8.** (**a**) Relationship of the fine tuning current and the output wavelengths. The injection current adjustment step size is 1 mA. (**b**) Wavelength dependence on temperature at a fixed current of 64 mA. The temperature change step size is 0.01 °C.

To determine the long-term stability of the IF-ECDL, we used a wavelength meter (WS7; HighFinesse) to monitor the frequency fluctuations of the laser during free running. The results obtained over a period of 18 h indicated good passive long-term stability, and The maximum deviation in laser frequency was only 200 MHz. Figure 9 shows the frequency stability of the ECDL derived from the Allan deviation. The measurements were conducted in an air-conditioned laboratory in which the temperature fluctuated by roughly 1 °C about an average of roughly 22 °C.

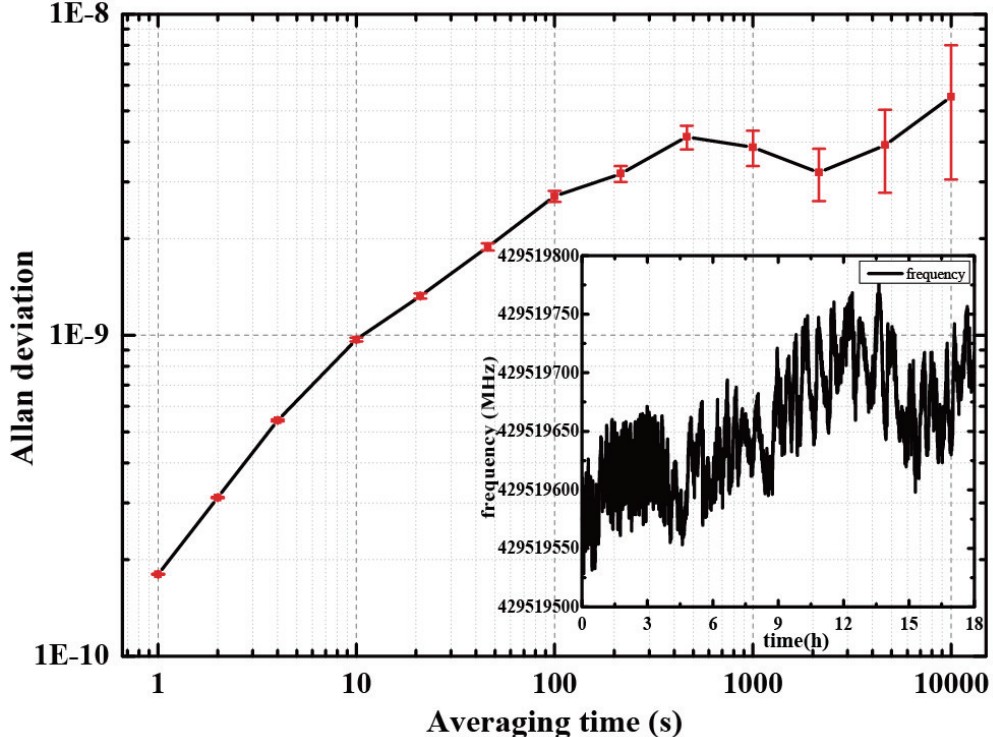

**Figure 9.** Long-term frequency fluctuations of the free-running 698 nm ECDL.

To determine the linewidth of the ECDL, we performed an optical heterodyne beat experiment involving the IF-ECDL and a 698 nm ultra-stable laser with an ultra-narrow linewidth. The spectrum of the beat signal is shown in Figure 10b. The full width at half maximum of the Lorentzian fit is 180 kHz. The linewidth of the ultra-stable laser is only ~1 Hz [26], and this was obtained by locking the laser frequency to a high-finesse optical reference cavity by means of the Pound–Drever–Hall method [22]. The linewidth of the beat signal can be considered to be the linewidth of the IF-ECDL because the latter is far wider than the linewidth of the ultra-stable laser. Therefore, the linewidth of the ECDL is roughly 180 kHz.

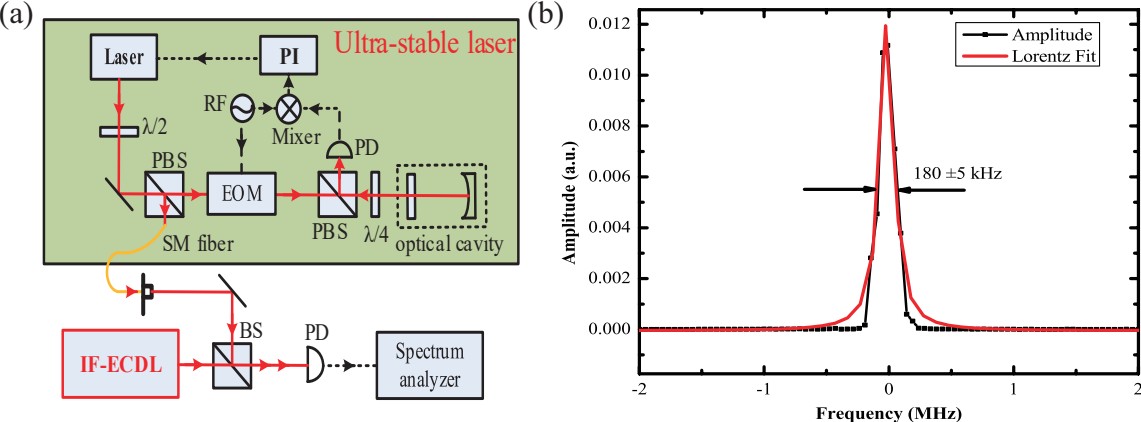

**Figure 10.** (**a**) Beat experiment involving the IF-ECDL and an ultra-stable laser with the linewidth of about 1 Hz. (**b**) Spectrum of beat signal between ECDL and a 698 nm laser with ultra-narrow linewidth. The injection current of the laser during the measurement is 64 mA, and The temperature is 22.3 °C. The resolution bandwidth of the spectrum analyzer is 10 kHz, and The sweep time is 1.29 s. The black line shows the power spectrum of the beat signal, and The red line indicates a fitted line.

## 5. Conclusions

In summary, this paper presents the design of a compact and robust ECDL for space applications. This ECDL was created without using any position adjusters, taking advantage of insensitivity to misalignment. As a wavelength-selective element, the laser uses an IF rather than a diffraction grating. The frequency of the first lateral rocking eigenmode is 2316 Hz. The ECDL emits 35 mW of laser power at a wavelength of 698 nm with a linewidth of around 180 kHz. In future work, we will conduct an adaptive test of the mechanical and thermal environment of the ECDL and optimize the design to make it more suitable for use in space.

**Author Contributions:** Conceptualization, T.L. and S.Z.; methodology, L.Z. and L.C.; validation, L.Z., L.C., G.X., and C.J.; writing—original draft preparation, L.Z.; writing—review and editing, L.Z., C.J., and J.L.; supervision, T.L. All authors have read and agreed to the published version of the manuscript.

**Funding:** This research was funded by Major Scientific Instruments and National Development Funding Projects of China (61127901, 91636101) and the Young Scientists Fund of the National Natural Science Foundation of China (11403031). The project was supported by the Open Research Fund of State Key Laboratory of Transient Optics and Photonics (SKLST201909).

**Conflicts of Interest:** The authors declare that there is no conflict of interest.

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
