# Peer review of "Development of an Interference Filter-Stabilized External-Cavity Diode Laser for Space Applications"

_photonics, doi:10.3390/photonics7010012_

Round 1

Reviewer 1 Report

The authors present a compact, narrow linewidth, red-emitting external cavity laser suitable for experiments in space. The paper is suitable for publication provided it is revised properly.

Provide the values of the reflectivities of both facets and the cavity length of the laser diode Include in Fig. 2 a frequency axis, too (for example at the top abscissa). Fig. 8 is difficult to grasp because of the 2 lines corresponding to wavelength and frequency. Please include only one line, but modify the right-hand ordinate accordingly (it is then reversed and not equally spaced) The authors write that the FSR is 3 GHz. But in Fig. 8 the modes jump over 40 GHz. Also the wavelength does not vary continuously between the mode jumps, in particular in dependence on the temperature. The authors should discuss these issues in detail in the paper. Fig. 10: Enhance the resolution of the Lorentzian fit. It seems to have too less points so that the top is extremely sharp. Give the injection current, the temperature and the measuring time for the linewidth determination. Is the linewidth achieved sufficiently small to probe the clock transition of strontium atoms?

Reviewer 2 Report

This paper describes a new design (without mechanical moving parts) for broadband near-infrared EC diode lasers for industrial and scientific applications.

Comments:

Abstract: Please indicate stability and tuning capabilities compared to the state-of-the-art or commercial lasers in this (698 nm) spectral range.

Line 4: Please change EDCL to ECDL

Introduction:

Line 19: Please clarify statement, “diode lasers to achieve linewidth and wavelength tuning”.

Line 26-27: Please re-write statement to indicate maximum wavelength and the incidence angle.

Line 32 – 46: Please re-write the entire paragraph, it is very difficult to read. If the importance of this work is to probe electronic transitions of Sr, please illustrate basic transitions, energy difference, or linewidth. This is will helpful when comparing laser linewidth and spectral resolution of the proposed ECDL.

Section2: Briefly explain working principle of interference filter to determine output transmission and wavelength selection.

Authors claim in the abstract the ECDL does not contain any “position adjustors” therefore, their proposed design is more stable. However, in the proposed research, the angle of the IF filter will be used to tune the output wavelength, how is this an “alignment immune” approach?

Section 3:

Fig.2: The angle of the IF is critical to determine wavelengths and tuning characteristics of the ECDL. Figure, 2 is data from the manufacturer which should have been re-validated by the authors. Please illustrate, uncertainties in transmission curves and linewidths due to error in angular placements of IF.

Line 137-147: Please illustrate impact of gravitational environment and the stress factors on the IF configuration, stability, which is crucial for this design. Results could be additional information provided in Figure, 6.

Figure 7: It is not clear how the amplification of laser power is achieved in the ECDL compared to a laser diode. Is the beam (or collimated beam) profile and M2 comparable in both cases?

General Comment:

There should be a description in the paper on background research, and current’ state of EC diode lasers and how this research advances the field. A brief discussion that compares frequency and mechanical stability and potential applications will strengthen the paper.

Round 2

Reviewer 1 Report

Unfortunately the authors did not respond to my remarks. Partially they were considered in the manuscript. But there are new issues raised.

Provide the values of the reflectivities of both facets and the cavity length of the laser diode The authors write that the FSR is 3 GHZ. But in Fig. 8 the modes jump over 40 GHz. Also the wavelength does not vary continuously between the mode jumps, in particular in dependence on the temperature. The authors should discuss these issues in much more detail in the paper.

Authors included not the length of the laser diode in the manuscript, did not improved the resolution of the Lorentzian Fit in Fig. 10 (there a too less points) and did not include the explanation for the frequency of the mode jump in the manuscript. Fig. 10 Enhance the resolution of the Lorentzian fit. It seems to have too less points so that the top is extremely sharp.

The statement starting at line 192 of the revised paper “(1) adding an external cavity is equivalent to an increase in cavity length; (2) introducing optical feedback to help increase the stimulated emission and suppresses the spontaneous” is not the correct explanation. because there "L" instead of "D" must enter. The threshold current is independent on the length of the external cavity as long as it is lossless (as here considered). The correct answer is, that the power is increased due to the decrease of the threshold current (as correctly written), and the threshold current is decreased due to the additional feedback from the external mirror with reflectivity R3, which decreases the outcoupling (or mirror) losses.

The statement starting at line 208 of the revised paper “This is because the numbers of electrons and holes in the active layer of the laser diode increased ….” is wrong. because when increasing the injection current, the temperature of the diode is increased, which results in an increase of the effective index (the number of electrons and holes increases only slightly).

Reviewer 2 Report

The abstract can further be modified for style and grammar. 

Author Response

Point 1: The abstract can further be modified for style and grammar.

Response 1: Thanks to the reviewer for this comment, and we have revised the abstract in the manuscript.